# Methyltransferases of *Riboviria*

**DOI:** 10.3390/biom12091247

**Published:** 2022-09-06

**Authors:** Arcady Mushegian

**Affiliations:** Division of Molecular and Cellular Biosciences, National Science Foundation, 2415 Eisenhower Ave., Alexandria, VA 22314, USA; mushegian2@gmail.com

**Keywords:** *Riboviria*, methyltransferase, S-adenosylmethionine, Rossmann fold, FtsJ, RrmJ, coronavirus, SN2 nucleophilic substitution, tobacco mosaic virus

## Abstract

Many viruses from the realm *Riboviria* infecting eukaryotic hosts encode protein domains with sequence similarity to S-adenosylmethionine-dependent methyltransferases. These protein domains are thought to be involved in methylation of the 5′-terminal cap structures in virus mRNAs. Some methyltransferase-like domains of *Riboviria* are homologous to the widespread cellular FtsJ/RrmJ-like methyltransferases involved in modification of cellular RNAs; other methyltransferases, found in a subset of positive-strand RNA viruses, have been assigned to a separate “Sindbis-like” family; and coronavirus-specific Nsp13/14-like methyltransferases appeared to be different from both those classes. The representative structures of proteins from all three groups belong to a specific variety of the Rossmann fold with a seven-stranded β-sheet, but it was unclear whether this structural similarity extends to the level of conserved sequence signatures. Here I survey methyltransferases in *Riboviria* and derive a joint sequence alignment model that covers all groups of virus methyltransferases and subsumes the previously defined conserved sequence motifs. Analysis of the spatial structures indicates that two highly conserved residues, a lysine and an aspartate, frequently contact a water molecule, which is located in the enzyme active center next to the methyl group of S-adenosylmethionine cofactor and could play a key role in the catalytic mechanism of the enzyme. Phylogenetic evidence indicates a likely origin of all methyltransferases of *Riboviria* from cellular RrmJ-like enzymes and their rapid divergence with infrequent horizontal transfer between distantly related viruses.

## 1. Introduction

Ivanovsky has shown in 1892 that the plant disease already known as tobacco mosaic is caused by a small infectious entity [1]. The entity was later named tobacco mosaic virus (TMV) and then assigned to the namesake species (family *Virgaviridae*). Much more recently it became known that one of the earliest events in the course of TMV infection is the presentation of a viral 5′-terminal guanosine (known as cap0) to the infected cell; this terminal structure is monomethylated in the N7 position and attached via an inverted 5′-5′ linkage to the genomic positive-sense RNA of the virus [2,3,4]. When exposed, this structure may become available to interact with cellular components, including ribosomes, before complete uncoating of the virions [5,6,7,8]. Moreover, the first protein domain that emerges from a ribosome upon translation of the genomic mRNA is the N-terminal domain of the 123/186 kDa non-structural protein of TMV, encoding a putative methyltransferase that is thought to generate cap0 [9,10]. Thus, unbeknownst to Ivanovsky, methylation of virus RNA is at the leading edge of virus interaction with the host.

The interest in biochemical modifications of mRNAs emerged nearly 60 years ago at the confluence of several research inquiries. In one line of work, mRNAs of viruses with DNA and RNA genomes were shown in the early 1970s to contain a non-templated methylated guanosine residue at the 5′-end, which led to the finding that a methylated cap attached via an inverted 5′-5-linkage is a general property of mRNA and some other classes of cellular RNA [2,11,12,13,14,15]. Ironically, TMV RNA was initially thought to be an exception based on the results of in vivo labeling [16], until the presence of cap in the virion RNA was established [2,3]. Around the same time, it was found that the growth of Sindbis virus in cultured insect cells depends on methionine added in the media, by way of methionine incorporation into S-adenosylmethionine (SAM) and further transfer of the methyl group onto the 5′-end of genomic RNA [17,18]. The mutation that conferred resistance to methionine depletion was mapped to a specific region of the non-structural protein encoded by the virus [19,20]. Review of that and other early work can be found in [15].

The region of the Sindbis virus non-structural protein that controlled efficient cap methylation showed amino acid sequence similarities to the proteins encoded by other viruses. An early report of sequence conservation from three groups of plant viruses and an insect-infecting Sindbis virus has speculated that some of the conserved regions may be involved in the formation of the cap [21], and later more detailed analysis defined the “Sindbis-like” family of virus SAM-dependent methyltransferases [9]. That family included domains from positive-strand RNA viruses such as alphaviruses, tobamoviruses, tobraviruses, hordeiviruses, tricornaviruses, furoviruses, hepatitis E virus, tymoviruses, potexviruses and carlaviruses [9]. In parallel, genome sequencing of RNA viruses from other groups, such as positive-strand flaviviruses and double-strand reoviruses, revealed similarity between distinct protein domains encoded by those viruses and cellular SAM-dependent methyltransferases involved in small molecule metabolism [22]. These putative virus methyltransferases were later found to be even more similar to another group of conserved cellular methyltransferases, typified by stress-inducible methyltransferases FtsJ [23,24,25,26,27], which modify 2′-hydroxyl groups of ribose in specific positions of rRNA and tRNA, and were in the meantime renamed RrmE/RrmJ [28].

The FtsJ-like/RrmJ-like methyltransferases (NCBI COG1189 and COG0893) are encoded in many groups of RNA viruses and are widespread in Bacteria, Archaea and Eukarya, and their biochemical activity has been documented for many representatives (see below). The RrmJ-like methyltransferases did not appear to be clearly related to Sindbis-like methyltransferases at the sequence level. More recently, it was found that the genomes of betacoronaviruses, including SARS-CoV and SARS-CoV2, encode a recognizable RrmJ-like domain (protein Nsp16, the most C-terminal domain in the large non-structural protein) and also the second methyltransferase Nsp13/Nsp14, which is not detected by typical similarity searches with the known methyltransferase sequences [29,30]. In addition, structural biologists studying the NS2 protease of alphaviruses serendipitously have discovered a fused domain topologically similar to methyltransferases, but not obviously related in sequence to the previously known methyltransferases ([31] and see below).

Besides the common cap methylation on the N7 atom of the base of the 5′-5′ linked guanosine (the cap0 event), cellular and virus mRNAs may be further methylated at the same guanosine or at the first few transcribed residues. Most 5′ ends of eukaryotic mRNAs are methylated at the ribose 2′-O positions in the first one or two transcribed nucleosides (known, respectively, as cap1 and cap2 modifications). In cellular snRNA and snoRNA, the cap0 guanosine is methylated twice more in the N2 position of the base. Examples of even more profound modifications of cap structure are also known, extending for example to four bases in trypanosome mRNAs [4].

In several viruses of the *Riboviria* realm, not only cap0 but also cap1 modifications have been documented. Viruses that encode two methyltransferases, such as reoviruses and coronaviruses, appear to use one enzyme to perform the cap0 methylation and the other for the cap1 event; on the other hand, in flaviviruses, the single enzyme called NS5 appears to perform both modifications [4,32].

Comparative sequence analysis has been used to detect conserved amino acid motifs in virus-encoded and cellular methyltransferases. One of the key studies was the analysis of sequence conservation among the SAM-dependent DNA methyltransferases involved in the DNA restriction-modification systems in bacteria [33]. The main result was the delineation of nine regions of high sequence similarity, designated Motif X and Motifs I-VIII (candidate Motif IX turned out not to be conserved when more sequences were added to the alignment). Though these motifs defined a family of DNA modification enzymes, these sequence motifs, sometimes in a modified form, are observed in all methyltransferases of *Riboviria* known thus far, as I will explain below.

Structural studies have established that many known cellular methyltransferases, as well as all methyltransferase domains encoded by *Riboviria*, belong to a Rossmann-like spatial fold, i.e., they are formed by β-strands and α-helices that alternate along the sequence, with the strands forming one central β-sheet and helices arranged into two layers, one on each side of the sheet. In Rossmann-fold methyltransferases, the most N-terminal β-strand is located in the middle of the sheet, and the sheet is filled from the inside out, to give the strand order 3214576. One of the hallmarks of many Rossmann-fold methyltransferases involved in DNA and RNA methylation is strand 7, which is antiparallel to all other strands and is not preceded by a helix, forming a β-hairpin with strand 6 [34]. Variations on this theme—most commonly replacement of one or more helices with long loops, or sometimes insertion of extra strands or helices—are observed in many virus-encoded methyltransferases. Such differences led some authors to propose that coronavirus Nsp13/14 does not adopt Rossmann fold at all [35,36], but such a suggestion is clearly refuted by the evolutionary classifications of structural domains, such the one provided in the ECOD database [37,38] and by the present analysis (see below).

A typical feature of Rossmanoid enzymes is that many functionally important, conserved residues can be found at the C-termini of the β-strands or in the adjoining loops. This was evident already in the early analysis of DNA methyltransferases by Malone, Blumenthal and Cheng (MBC in the rest of this paper; [33])—when compared to the three-dimensional structures of two DNA methylases known at the time, motifs I, II, IV, IV and VIII corresponded to the β-strands and adjoining loops, whereas motifs X, III, V and VII mapped to α-helices and were less well conserved. All motifs except Motif VII were found on the concave surface of the molecule, and many were implicated in specific interactions with the ligand SAM or with a polynucleotide substrate; Motifs X (the region closest to the protein N-terminus) and I–IV (following Motif X along the protein sequence) were mostly involved in the interactions with SAM, and motifs V–VIII had roles in interacting with both SAM and the substrate [33]. These general observations were confirmed and extended by the structural analysis of many RrmJ-like methyltransferases from *Riboviria*, as discussed below.

Despite all the knowledge about sequences, structures and functions of many SAM-dependent methyltransferases encoded by RNA viruses, there was no clarity about the evolutionary relationships between RrmJ-like and Sindbis-like methyltransferases (nor about the sequence affinities more derived second alphavirus methyltransferase-like domain and coronavirus Nsp13/14). Do all these methyltransferases adopt a particular version of the Rossmann fold because they have diverged from a common ancestor that had that structure, or because of parallel or convergent evolution from dissimilar sequences and structures, towards similar molecular function? If the common ancestry hypothesis is true, we may expect that in addition to the same spatial organization, most virus methyltransferases would share some common sequence motifs. This would have implications for better recognition of novel virus methyltransferases, for understanding of the methyltransferase catalytic mechanisms, for engineering the enzymes with desired properties and for devising strategies for anti-virus defense.

In this article, I present the evidence that all RNA methyltransferases of *Riboviria* are related at the sequence level, preserving their distinct versions of at least eight out of nine motifs first delineated in bacterial DNA methyltransferases. A well-known sequence signature K-D-K-E in the RrmJ-like methyltransferases identified by Feder et al. (FPBW in the in the rest of this paper; [39]) turns out to be a specific realization of four of these shared sequence motifs. The alignment generated here, and the probabilistic models derived from it, may be useful for detecting putative methyltransferase domains in newly sequenced virus genomes. 

## 2. Materials and Methods

Virus taxonomy 2021 release (https://ictv.global/taxonomy/history, accessed on 27 October 2021) was used to select a representative member of each family in the realm *Riboviria*. As a rule, a species with completely sequenced genome was randomly selected from GenBank Genome Division, though occasionally a sequence of a domain from the same family with the known three-dimensional structure was included instead. For all proteins encoded by each selected virus, their annotations in GenBank were examined, and the presence of any domain recognized as SAM-dependent methyltransferase was recorded. 

The amino acid sequences of the GenBank-annotated methyltransferases from *Riboviria* were compared using an iterative approach, alternating the automated multiple alignment step with interactive homology-based re-definition of the domain boundaries and re-alignment of internal regions. The multiple alignments were obtained using the PROMALS-3D server [40]. The re-definition of domain boundaries was done by visual inspection of conserved sequence motifs within the PROMALS-3D alignment, addition of obviously missing terminal regions to the domains defined in GenBank, and occasional adjustment of clearly misaligned internal regions. The programs PSI-BLAST [41] and HHPred [42] were used throughout the analysis to test for the presence of additional conserved regions. PSI-BLAST was run with the following parameters: composition-based statistics: ON, E-value inclusion cutoff: 0.05, the number of sequences to report: 10,000; HHPred was run with the parameters: MSA generation method: PSI-BLAST, Alignment_mode: each of the three modes in turn (i.e., local_norealign, local_realign, global_realign); other parameters in both programs were left at the default values. Throughout the study, the FASTA database of all proteins from *Riboviria* with completely sequenced genomes, downloaded from NCBI Genome Division, was used as the search space (62,200 sequences with 38,195,545 combined residues as of September 2020).

Comparison of the three-dimensional structures was done using the Dali server [43]. The phylogenetic tree of representative methyltransferases of Riboviria was produced using the PhyML algorithm implemented through the BOOSTer service (booster.pasteur.fr). The statistical support for the tree partitions was assessed with bootstrap-by-transfer approach [44]. Phylogenies were visualized using the iTOL environment [45]. The three-dimensional structures of proteins were examined and visualized using the open-source PyMOL environment ([46]; SciCrunch RRID SCR_000305).

## 3. Results

### 3.1. The Search for Methyltransferases Encoded by Riboviria, Their Multiple Alignment and Identification of Conserved Sequence Motifs

I selected a representative member of each family in the realm *Riboviria* and identified all domains recognized as SAM-dependent methyltransferases. The list of viruses collected at that stage is given as Appendix A. If methyltransferase domain was not annotated, other viruses with completely sequenced genomes from the same family were examined, and if found, the domain was included in the starting dataset. If a methyltransferase domain in a given family has been documented in the literature, but none of the members had a relevant GenBank annotation, no family representative was included at first, but they were recovered by similarity searches at the next steps and added to the model.

Sometimes only a partial match to a known methyltransferase domain is recorded in GenBank; in such cases, the adjoining portions of the sequence were examined with the PSI-BLAST and HHPred programs to account for the conserved regions that were missed by the GenBank annotations. PSI-BLAST program and HMMer3.0 package were also used to search the database of all proteins from *Riboviria* with completely sequenced genomes with each of the sequences as a query. This approach was useful for defining methyltransferase domains in those virus families that have not yet acquired GenBank annotations. The multiple alignments were then obtained using the PROMALS-3D server that utilizes the information on the homologs of each sequence in the database and the evidence from the known three-dimensional structures. The domain boundaries were re-defined interactively as explained in Methods. I performed with 5 cycles of such iterative search and realignment, recovering many unannotated methyltransferase domains.

As a rule, a member of each of the groups of methyltransferases discussed above (i.e., Sindbis-like, RrmJ-like, coronavirus 13/14-like and alphavirus protease-fused methyltransferases) detected only members of the same group in the probabilistic sequence searches using the alignment models. No high-scoring false positives were encountered in any of the searches, with the exception of an occasional match to the Rossmann-fold helicase domains, mostly from flaviviruses, which were, however, easily identified as false positives by the examination of the signature sequence motifs. A stronger support of homologous relationships between all four families was provided by a structural comparison of representatives of all groups of methyltransferases mentioned above to the entire PDB database. The results of analysis with the Dali algorithm revealed close structural similarities between methyltransferases of *Riboviria*, capping enzymes of DNA viruses and eukaryotes, and FtsJ/RrmJ family of ubiquitous RNA methyltransferases. In the vast majority of cases, a match between an RNA virus methyltransferase from each group and a cellular FtsJ homolog had Z scores between 12 and 14, and the root mean square deviations (RMSD) of superimposed domains were between 2.5 and 3.5 angstrom. Matches between different virus methyltransferases were typically in the same value range. One exception was the Nsp14 protein of coronaviruses; in this case, the top non-self match for the representative structure (PDB ID 5c8s, Chain B) was to the cellular capping enzyme (PDB ID 1ri5), with Z score 8 and RMSD 3.0 angstrom. In those searches again, all other significant matches were only to the known SAM-dependent methyltransferases.

Notwithstanding the spread of the Z scores and RMSD values, all of them are within the range that indicates likely homology between the sequences whose structure is compared [37,47,48]. The same conclusion can be drawn from the evolutionary classification of sequences and structures obtained by a consensus of multiple methods and captured in the ECOD database. There, multiple structural families of Rossmann-fold SAM-dependent methyltransferases of *Riboviria* are joined into a single Topology group (ECOD 5 January 2003), which strongly suggests their common evolutionary origin [37,38].

One way to strengthen the above observations even further is to find out whether the entire set of methyltransferases of *Riboviria*, and possibly the cap methylating enzymes of cells and DNA viruses, as well as RrmJ-like RNA methyltransferases, share not only the spatial folds but also have the common sequence elements. To study this question, I used an iterative approach, alternating automated alignment with the PROMALS-3D program and manual inspection of the indel placement and missing portions at the ends of the alignment. At each iteration, progressively more diverse sequences were added to the alignment. 

At the last step, the alignment was converted into a Hidden Markov Model, and the database of all proteins from *Riboviria* with completely sequenced genomes was searched with the hmmsearch program of the HMMER3 suite, with a permissive parameter set (sequence reporting threshold E-value 50, domain reporting threshold E-value 50, sequence inclusion threshold E-value 10). There were no false positives apart from the occasional Rossmann-fold helicase domains mentioned above, and additional true positive matches from virus families that have not been included into the query were added to the multiple alignment as described above. The final alignment produced after five iterations analysis is shown in Figure 1.

The main result is that nearly all motifs identified by MBC are found in almost all of the methyltransferases analyzed here. Only the region surrounding Motif III could not be unequivocally aligned, whereas Motifs X, I, II and IV-VIII are clearly recognizable at the sequence level and are further supported by similar location within the known or predicted secondary and tertiary structure elements (Figure 1). Moreover, the K-D-K-E signature defined for a subset of FtsJ-like methyltransferases by FPWB [39] is clearly visible in all *Riboviria*, with the first and second of these hallmark residues particularly well conserved within, respectively, Motif X and Motif IV. The third and fourth of the signature residues (K in Motif VI and E in Motif VIII) are also conserved in the majority of the sequences, and are replaced mostly in methyltransferases from the order *Tymovirales* (Figure 1).

The alignment shown in Figure 1 and the Hidden Markov Model based on it (available as Appendix A) may be useful for annotation and verification of SAM-dependent methyltransferases with Rossmann fold in the newly sequenced genomes of *Riboviria*. I also tested whether *Riboviria* may encode other classes of methyltransferases, such as SPOUT family (a distinct Rossmann-fold methyltransferase that has a different set of conserved sequence motifs [49]) and SET family (an unrelated beta-clip fold [50]). Hidden Markov Models prepared from the alignments of those families from the Interpro database [51] were used to scan the *Riboviria* proteins with the hmmsearch program, but there were no matches.

Either one or two Rossmann-fold SAM-dependent methyltransferase domains were detected in all virus families where such an activity and/or the presence of a methylated cap structure have been known or postulated before. There were only two exceptions. First, a putative methyltransferase domain has been proposed in rubella virus (*Rubivirus rubella*; *Matonaviridae*), though it has been noted that the fit to the consensus pattern was poor [9]. I was unable to identify a methyltransferase homology region among the non-structural proteins of rubiviruses, and the HHPred prediction of secondary structure in the candidate region suggested scarcity of β-strands; it would be interesting to investigate whether rubella virus genome encodes a modified or a novel methyltransferase, dissimilar from the methyltransferases of its nearest evolutionary neighbors, hepeviruses and beneviruses. Second, most families within the order *Mononegavirales* encode for a well-defined RrmJ-like methyltransferases in the C-terminal regions of their large non-structural proteins. Viruses within the family *Bornaviridae*, however, do not appear to have a homologous domain.

### 3.2. Functional Roles of the Conserved Motifs and the Mechanism of Methyltransferase Reaction

Structural biologists have obtained many high-resolution spatial structures of methyltransferases from *Riboviria*. I examined these structures, paying particular attention to the more recent contributions that include co-crystallized native ligand S-adenosylmethionine. The known functions and interactions of the conserved sequence motifs are briefly summarized below. It should be noted that the architecture described in this section abstracts away many details—most virus methyltransferases have insertions and deletions of variable length, and additional functionality provided by protein cofactors of the main methyltransferase. These complexities, important as they are, are not considered here, in an attempt to focus on the universal properties of all or most methyltransferases in *Riboviria*.

Motif X consists of a long α-helix with a highly conserved lysine residue in the middle, rarely replaced by an arginine. This residue is the first K in the K-D-K-E signature. The side chain of this lysine is frequently involved in the network of hydrogen bonds with conserved residues in other motifs, and when water is explicitly placed in the structure, these residues also can be found in contact with one or two water molecules that are part of the same network. The significance of this will be discussed below.

The next conserved sequence block, Motif I, consists of a β-strand (strand 1 in the conserved core of the Rossmann fold) followed by a glycine-rich loop. In some viruses the sequence in this loop fits a consensus expression GxGxG, though in general the configuration of the small-chain, bend- or kink-prone residues in this location is more fluid, especially in *Tymovirales* (Figure 1). The loop connects the β-strand to an α-helix. In all known structures, one or more residues in the loop contact the ligand, typically its carboxypropyl moiety. Unusually for a β-strand, in its middle there is a well-conserved charged residue, most commonly an aspartic or glutamic acid, sometimes replaced by a tyrosine or a histidine. The essential role of this residue for capping and virus infection has been shown for the Sendai virus methyltransferase [52], but its precise molecular function remains unclear; it is apparently also part of a dense hydrogen bond network in the ligand-binding site, but does not seem to make direct contacts with SAM.

Motif II consists of β-strand 2 and adjoining turn. An aspartic acid residue at the C-terminus of the strand is seen often, though it may be substituted by an asparagine or sometimes glutamic acid. In the known structures this residue tends to form hydrogen bonds with the hydroxyls of the ribose within SAM. More variable residues within this strand may interact with the adenosyl in the ligand. 

Motif IV includes β-strand 4, which is found at the edge of the β-sheet in the Rossmann fold. An aspartic acid residue is highly conserved at the C-terminus of this strand, replaced only infrequently by glutamic acid. This is the D of the K-D-K-E signature, which sometimes makes contact with the nitrous base of the ligand, and, similarly to the K in motif I, is seen to coordinate a water molecule, whenever the solvent is included in the structure. This aspartic acid residue is almost always essential for the methyltransferase activity

Motifs III, V and VII of MBC are connector helices (with added strand 3 in Motif III) that are likely to determine the proper positioning of the main β-strands and the structural integrity of the core; no strong patterns could be identified in these elements, especially in the hypervariable helix in Motif III. 

Motif VI consists of β-strand 5, preceded by another glycine-rich loop. Within this motif, a lysine residue (the second K of the K-D-K-E pattern) is often found, though it is poorly defined in *Tymovirales*. Residues in the vicinity of Motif IV often contact the amino group and sulfonium group of the methionine within SAM. The K is not always essential for methyltransferase function.

Motif VIII comprises a β-hairpin formed by strands 6 and 7. The loop that connects the two strands is serine-rich in FtsJ-like methyltransferases. A glutamic acid residue (E in K-D-K-E) is found in a nearly-invariant position at the N-terminus of strand 7 in most sequences. As with Motifs I and VI, however, this site is remodeled in *Tymovirales*. Interestingly, despite such a sequence variability in this and two other conserved motifs, the overall sequence similarity between *Tymovirales* proteins and other Sindbis-like virus methyltransferase is high enough to be confidently detected by the standard similarity search programs such as PSI-BLAST and HHpred.

Many of the functional implications of the structural information summarized above have been thoroughly tested experimentally, and the wealth of information about molecular and biological phenotypes of various mutations in the conserved and variable residues in many virus methyltransferases is available. Moreover, the interest in understanding the mechanism of action of antivirus compounds, and the need to develop novel antivirals, has resulted in structural and functional studies of methyltransferase interactions with small-molecule inhibitors. Studies of kinetics of ligand, cofactor and substrate binding and catalysis are also available. I cannot possibly do justice for this extensive body of knowledge by briefly reviewing it here, and defer to the primary studies and reviews that point towards a broader context [25,27,32,53,54,55,56,57,58,59,60,61,62,63,64]. It is, however, of interest to see whether the picture of sequence conservation delineated above may help to answer a more fundamental question of the details of the chemical mechanism of methyl transfer from SAM to a nucleotide.

The reaction of methyl transfer using the SAM ligand/donor proceeds by the equation R-H + 5′-((3-amino-3-carboxypropyl)methylsulfonio)-adenosine [*S*-adenosyl-L-methionine] = R-CH3 + 5′-*S*-(3-Amino-3-carboxypropyl)-5′-thioadenosine [*S*-adenosyl-L-homocysteine]
where R in the cases before us is an amido- or oxo-group of a nucleotide base or sugar moiety in the virus RNA. The transfer of the methyl group from SAM to the acceptor is thought to occur by a nucleophilic substitution of the SN2 type. In the early days of the studies of RNA methylation, however, this did not seem self-evident, as the chemistry of nucleoside derivatives in vitro strongly suggested that nucleobase methylation may proceed via a non-intuitive Dimroth rearrangement in the ring [65]. Experiments that involved radioactive labeling of atoms in specific positions of the base excluded this possibility [65], and the detection of the inverted configuration in the chiral methyl group in a uridine methylation within a tRNA strongly suggested that the transfer may take place by a simple SN2 displacement mechanism [66]. The leaving group in the SN2 reaction is quite obviously *S*-adenosylhomocysteine, but the identity of the nucleophile has not been determined for any RNA methylation reaction for a long time. In a 2005 communication [34], I wondered whether the conserved charged residue in Motif I could play such a role, but the inspection of the spatial structures of the virus enzymes does not appear to support this possibility—the orientation of that residue seems to be off.

More recently, structural studies indicated that none of the conserved potentially nucleophilic residues in virus methyltransferases would be positioned at the appropriately short distance in a correct orientation behind the methyl group of SAM in the enzyme active centers; in fact, it has been argued that the best candidate for a nucleophile is the substrate itself, or more precisely, the deprotonated O or N atom that is to be modified, e.g., [53,55,56,67]. In this framework, the conserved core of the enzyme and the invariant polar residues perform their function by correctly orienting the ligand and the substrate, and possibly also by helping to deprotonize the amine or hydroxyl group of the substrate, which initiates the reaction.

While analyzing the three-dimensional structures of methyltransferases in the light of the sequence conservation shown in Figure 1, it became evident that, whenever the native ligand is co-crystallized with the protein and the solvent molecules are explicitly accounted for in the structure, a water molecule is seen at the vicinity of the methyl group of SAM. That molecule is coordinated by nearly-invariant residues, K in Motif X and D in Motif IV. Such an arrangement is seen in a cellular RrmJ-like enzyme and in RrmJ-like Nsp16 of coronaviruses and NS5 of flaviviruses (Figure 2). 

It is tempting to suggest that this water molecule plays a role in the catalytic mechanism. A radical proposition would be that this water, polarized by the charged residues in its environment, is in fact the entity that initiates a nucleophilic attack on the methyl group of SAM. Two main arguments against this idea are, first, that the other transfer step would still be needed to ligate the methyl group to the recipient (this would make the reaction mechanism, strictly speaking, something other than SN2), and, second, that water as a nucleophile has been implicated in hydrolase and lyase/isomerase reactions but is relatively uncommon in transferases ([68,69], but see [70]). Perhaps a more realistic possibility is that this water is needed to protonate the leaving group, i.e., *S*-adenosylhomocysteine. Very recently, such a possibility has been pointed out in the context of Nsp16-mediated catalysis; the picture there is complicated by the possible role of metal ions close to the active center, but the function of the water may be the same [67]. Be it as it may, the role of bound water molecules in methyl transfer warrants further investigation, especially for the methyltransferases outside of the RrmJ group that are not well studied from a structural point of view.

### 3.3. Gain and Loss of Methyltransferases in the Evolution of Riboviria 

The sequence and structure similarities described in this work suggest that all methyltransferases of *Riboviria* are homologous, i.e., that they have evolved from a common evolutionary ancestor. As far as I know, this has not been explicitly stated before, and the phylogeny of the entire set of RNA virus methyltransferases has not been analyzed. As a first approach to the problem, I used the alignment shown in Figure 1 to generate a maximum-likelihood phylogenetic tree, assessing validity of the internal partitions in the tree with the bootstrap-by-transfer approach. In simulations, that method is reported to have higher resolution than the traditional bootstrap and to generate fewer falsely supported branches [44]. Also included in the alignment and in the tree are representative sequences of mRNA capping enzymes encoded by cells and by viruses with DNA genomes, as well as RrmJ-like methyltransferases that modify cellular rRNA and tRNA. The methyltransferase tree topology in the Newick format and tree visualization are included as Appendix A. 

Given the large divergence between the sequences in the alignment, it is not unexpected to see that there are very few partitions in the tree whose topology is statistically supported. The existence of “Sindbis-like” and “RrmJ-like” clades of methyltransferases, however, can be stated with confidence—the split of the entire tree into those two partitions has the bootstrap-by-transfer values of 70%. Within the “Sindbis-like” partition, there is another well-supported clade, corresponding to the order *Tymovirales*. Also, all methyltransferases encoded by cells and a DNA virus are in the RrmJ-like part of the tree, and the cellular enzymes are seen as deep branches within that part. Revisiting Figure 1, the reader will notice that the aligned sequences are grouped according to the Sindbis/RrmJ split generated in the tree. At a higher resolution, the order in which the sequences are sorted within the groups in Figure 1 correspond to the raw order in which the tips of the tree provided in Appendix A can be traversed.

At the first glance, an evolutionary interpretation of all this seems difficult. However, when the presence, absence and identity of the methyltransferases in *Riboviria* is mapped onto the current phylogeny of the realm, a much more coherent picture transpires. That mapping, which uses the tree of the ubiquitous RNA-directed RNA polymerase domain as the reference [71], is shown in Figure 3.

The understanding of the *Riboviria* phylogeny that has emerged in the last 5–10 years identifies 5 main clades of the RNA virus world (see [71,72,73] for the in-depth discussion). Those clades (called Branches in the rest of this section, following [73]) are as follows. 

Branch I. Viruses with single-strand positive-sense RNA genomes that infect prokaryotes, and a small collection of related viruses of fungi and plants. 

Branch II. A complex, yet apparently monophyletic, assembly of single-strand positive-sense RNA viruses. Some of virus families in that branch have the “picorna-like” mode of genome expression, i.e., their 5′ end is adorned with a genome-linked protein VPg instead of a cap structure; protein synthesis is initiated at an internal site on RNA and results in making a giant polyprotein that is proteolytically processed into the functional modules. Yet other taxa within Branch II, such as families *Flaviviridae* and *Polymycoviridae*, as well as the order *Nidovirales*, have capped genomic mRNA and different modes of genome expression. 

Branch III. An expanded collection of viruses of eukaryotes with single-strand positive-sense RNA genomes that includes groups of viruses previously recognized as “Sindbis-like”, “toga-like” or “rubi-like”.

Branch IV. Viruses of prokaryotes and eukaryotes with double-strand RNA genomes—this group might be paraphyletic.

Branch V. Viruses of eukaryotes with single-strand negative-sense or ambisense RNA genomes.

In the light of this megataxonomy of *Riboviria*, we see the evolutionary logic in the diversity and distribution of methyltransferases within the realm (Figure 3). Viruses in Branch I infect mostly prokaryotes and do not methylate their RNA. Viruses in Branches II, IV and V rely on the RrmJ-like methyltransferases to modify their mRNAs, but they also exhibit frequent replacements of the de novo cap synthesis by other strategies of synthesizing their 5′-ends, including aforementioned VPg and internal initiation module, as well as cap-snatching from the cellular mRNAs in the ambisense-strand *Bunyavirales* and *Articulovirales*. And finally, viruses in Branch III have capped genomic mRNAs and rely almost exclusively on “Sindbis-like” methyltransferases to do so; one exception is the family *Tombusviridae*, which may have lost the cap and RNA methylation function secondarily (also *Tombusviridae* is a deep branch within this megaclade, and alternative placements, for example, within Megaclade II, have been suggested).

The following evolutionary scenario seems plausible in the light of these observations. Early in the evolution of eukaryotes, the cells developed a molecular module that modified the 5′-termini of their RNAs. The source of the methyltransferase activity within the module was a derived version of an RrmJ-like enzyme, that had already emerged and diversified within bacteria and archaea [24]. As *Riboviria* were adapting to the new cellular environment, they have acquired this trait too, possibly as part of molecular mimicry to evade the host systems of RNA surveillance (see Discussion). To do so, they either exapted a cellular cap-modifying enzyme or repurposed another similar RrmJ-like methyltransferase; in any case, a large swath of phylogenetically diverse viruses of today have retained the evolved but recognizable offspring of that ancestral methyltransferase of *Riboviria*. Only one group, the present-day Branch III, has experienced a rapid acceleration of the evolutionary rate in their methyltransferase domain, which produced a monophyletic, mostly synapomorphic trait of “Sindbis-like” methyltransferases within this branch.

Rare but interesting exceptions from this overall trend are also seen in Figure 3. In addition to already-discussed repeated losses of the cap methylation trait, the following evolutionary events may have occurred. First, in the plant viruses of the *Aspiviridae* family, the usual for Branch V RrmJ-like methyltransferase is replaced by a Sindbis-like paralog—conceivably as a result of a recombinational horizontal transfer from a Branch III virus. Second, genomes in four groups of viruses encode not one but two methyltransferases; this is true for order *Nidovirales* (Branch II), family *Kitavirales* in broad sense (i.e., including the negevirus group) and order *Togavirales* (both in Branch III), and family *Reoviridae* (Branch IV). In those cases, one of the methyltransferase domains in the same virus genome is from the RrmJ group and the other is from the Sindbis group (Figure 1). This could indicate that there have been up to four independent horizontal acquisitions of an additional methyltransferase activity in the history of *Riboviria*. At a more local scale, some within-family horizontal gene transfers of methyltransferases also may have occurred; accounting for such events requires a deeper examination.

## 4. Discussion

In this work, I argue that sequence and structure comparisons of the putative methyltransferase domains encoded by diverse *Riboviria* reveal the evolutionary signal that is sufficient to suggest the monophyletic origin of these domains. Multiple sequence alignment unifies many previously noticed conserved sequence motifs, and the pattern of sequence conservation in the (super)family suggests targets for exploring the chemical mechanism of methyl transfer and devising the ways to inhibit the enzyme activity.

With a narrow focus on the methyltransferase domains, this work does not address an important broader question of the diversity of cap biosynthesis pathways in *Riboviria*. Cap0 formation requires at least three steps, i.e., hydrolysis of the terminal triphosphate of a nascent mRNA to generate a diphosphate at the 5′-end, transfer of a guanine monophosphate nucleoside to the 5′-end, and methylation of the guanine at N7. Variations of this pathway are also known, such as the use of a guanine diphosphate instead of mRNA diphosphate as the source of the 5′-5′ triphosphate linkage, and the swap in the order of the last two stages, whereby guanine monophosphate is getting methylated before it is attached to RNA [4,10,74]. The protein domains and cofactors required for completing the pathway in vivo are likely to be different, even non-homologous, in different viruses. 

The census of methyltransferases presented here, when considered jointly with the biochemical studies, may refute an expectation that the number of methyltransferase domains within a virus proteome determines the number of cap methylation events. Indeed, a single methyltransferase of flaviviruses catalyzes both cap0 and cap1 events, whereas Sindbis virus, believed to have a cap0 structure [18], encodes two methyltransferases (Figure 1). To get a deeper insight into the cap methylation events, it may be timely to revisit the structure of 5′-termini in mRNAs of *Riboviria* with the new-generation methods for chemical characterization of RNA, and then match the set of modifications to the repertoire of the candidate enzymes. 

Surveys of modified residues in virus mRNAs have recently found methyl groups attached not only to the few 5′-terminal residues, but also to the nucleosides (the base or the sugar) in the mRNA body (recently reviewed in [75,76]). Neither the extent and site-specificity of those modifications, nor their biological functions have been sufficiently studied. That said, it does not seem to be out of question that some of virus methyltransferases could modify internal sites in virus genomes, either serendipitously or specifically. It is also notable that some plant viruses encode dealkylation/demethylation enzymes of the AlkB family, and the suggestions of the role of such enzymes vary from the repair of non-specific alkylation damage to the erasure of putative epigenetic marks in mRNA [77,78,79]. 

In the case of RNA viruses that infect animal cells, it is becoming clear that the methylation status of the RNA 5′ end plays a role not only in enabling efficient translation and reducing the rate of non-specific mRNA decay, but also in the virus-host arms race. In one branch of cell intrinsic immunity, the cap0 structure is recognized by the animal cells as a pathogen-associated molecular pattern by the RNA sensor RIG-I, and virus reproduction is suppressed through the pathway involving MDA5 and interferon-induced restriction factors such as IFIT1 [76,80,81,82,83,84]. Decrease in flavivirus and coronavirus 2′-O methylation has been shown to make virus reproduction sensitive to restriction by type I interferon; in cells where the IFN signaling or IFIT1 levels are knocked down, the replication of cap0 virus is restored [76,80,85]. Plants and fungi have no RIG-I or IFIT genes, lack the interferon response pathway, and the plant innate immunity sensors are believed to sense the presence of viruses not through virus RNA but through specific virus-host protein-protein interactions [86,87]. Given that plant viruses in at least two families, *Reoviridae* and *Kitaviridae*, have evolved two methyltrasferase domains, it would be interesting to know whether plants do in fact possess the RNA sensing modalities that are able to discriminate between cap0 and cap1 structures, which the second methyltransferase may help to avoid. It is also notable that the methyltransferase domain of TMV us able to reduce the extent of RNA silencing by the host [78,88]. A different kind of anti-virus defense mechanism, RNA-mediated gene silencing, depends on small RNAs and the proteins of Argonaute family in animals, plants, fungi and a subset of prokaryotes [89]. It is not fully understood whether these pathways may sense virus RNAs on the basis of their methylation status, but the links between cap-binding protein complexes and RNA silencing systems have been noted in plants and animals [90,91,92,93,94]. It would be interesting to know whether these methyltransferase-centered mechanisms of countering host resistance are responsible for some of the pathological states in the host, such as the visible symptoms that prompted Ivanovsky’s research on tobacco mosaic disease more than 130 years ago.

## Figures and Tables

**Figure 1 biomolecules-12-01247-f001:**
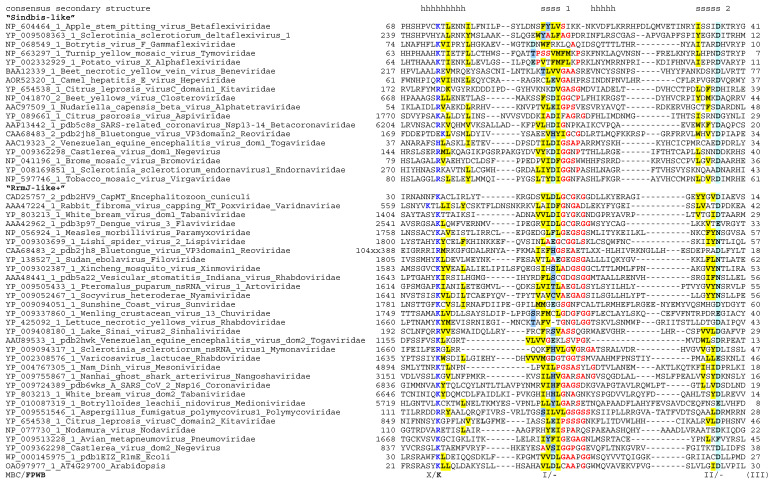
Multiple sequence alignment of SAM-dependent methyltransferases encoded by representative genomes of *Riboviria*. Sequence identifiers in GenBank and PDB, virus species names and the family names, as of 2021, are shown on the left (truncated in the lower pane). The distance from the N-termini of the proteins, in amino acids, is shown before each sequence, and the lengths of less-conserved inserts are also shown. In the consensus secondary structure lines above the alignment, s stands for a β-strand and h stands for an α-helix; the positions are marked if the residue is found in a strand or a helix in more than a half of the known spatial structures, and the numbering of the strands within the Rossmann fold is provided. Conserved hydrophobic residues (I, L, M, V, F, Y, W) are indicated by yellow highlight, conserved small-side-chain or turn/kink-prone residues (A, G, S, P) are indicated by bold type and red color, conserved positively charged residues (K, R) are in bold type and blue color, and conserved acidic or amine side chains (D, E, N, Q) are highlighted in cyan. In Motif I, blue highlight is used to mark additional residues with potential negative charge (C, H, S, T and Y). Prior motif nomenclature (see text) is shown under the alignment. The order in which the sequences are sorted is explained in the Section 3.3.

**Figure 2 biomolecules-12-01247-f002:**
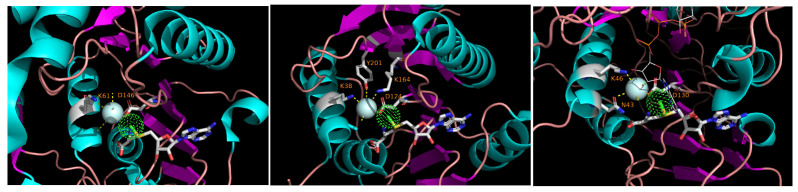
Bound water molecules in the active centers of diverse RrmJ-like methyltransferases. The cyan sphere represents water, and the green dotted sphere represents the methyl group of the bound S-adenosylmenthionine rendered as licorice sticks. The side chains of the residues that ligate the water molecule are shown and marked; the numbering is from the beginning of a pdb entry, not from the beginning of the virus (poly)protein as in Figure 1. (**Left**), NS5 methyltransferase of Dengue virus 3 (pdb 3P97); center, methyltransferase RmlE of *E. coli* (pdb 1EIZ); (**Right**), coronavirus methyltransferase Nsp16 (pdb 6WKS).

**Figure 3 biomolecules-12-01247-f003:**
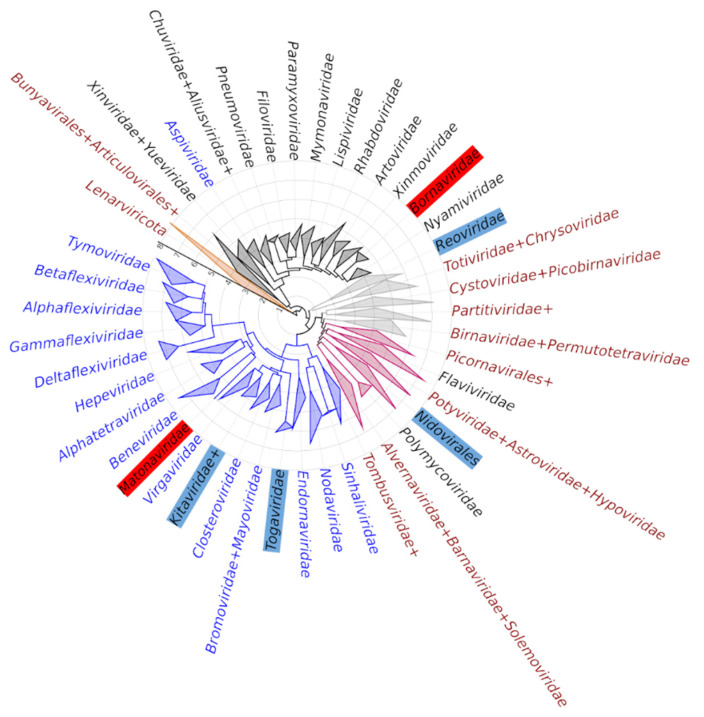
The distribution of methyltransferases among the families within the realm *Riboviria* tree. The tree is based on the most recent update of the megataxonomy of *Riboviria* and is constructed from the alignment of more than 6000 virus RNA-directed RNA polymerases (71); for the purpose of this visualization, the clades have been collapsed at the family level or occasionally at the order level. Some colors used for coding the clades (the inner circle) and the text labels (the outer circle) are slightly similar, but the coding schemes are different, and are as follows. The clades, counterclockwise from 10 o’clock: golden, Branch I; dark gray, Branch V; light gray, Branch IV; magenta, Branch II; blue, Branch III. The text labels: maroon, virus families that do not encode methyltransferases; dark gray, virus families encoding RrmJ-like methyltransferases; blue, virus families encoding Sindbis-like methyltransferases; black on blue, virus families encoding two different methyltransferases; black on red, unconfirmed methyltransferase domains in *Bornaviridae* and *Matonaviridae*. The concentric grid and the axis pointing at 10 o’clock indicate the tree scale, measured in the average number of substitutions per site.

## Data Availability

The data supporting reported results can be found as the Appendix A submitted with this manuscript.

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
