# Peer review of "Methyltransferases of Riboviria"

_biomolecules, 2022, doi:10.3390/biom12091247_

Round 1

Reviewer 1 Report

In this MS the author has explored very interesting features regarding the described RNA methyltransferases distributed across several members of the realm Riboviria and their homologues in some cellular counterparts. By analysis involving the comparison of the sequence motifs and predicted three-dimensional structure of these proteins, the author has observed a strong statistic support to suggest a common evolutionary origin that links the set of methyltransferases in members of this viral realm and the cellular widespread  RrmJ-like methyltransferase proteins. In addition, when further investigation was done and the three-dimensional structures of some viral and bacterial RrmJ-like proteins were analyzed, the author also noticed a molecule of water bound in the active center of these proteins. At the end, in light of his previous observations,  the author describe a plausible evolutionary scenario for the origin of methyltransferases in different viral families inside the Riboviria realm    page 01 - line 31: Please change "later" for "then".   page 01 - line 32-35: "Much more recently it became known that one of the earliest events in the course of TMV infection is presentation to an infected cell of the 5'-terminal guanosine, which is monomethylated in the N7 position and attached via the inverted 5'-5' linkage to the genomic positive-sense RNA of the virus. This terminal structure, known as cap0, may become exposed and available for interaction with the cell, including ribosomes, before complete uncoating of the virions". I would suggest a slight modification in the sentence for the sake of comprehension . Feel free to change it or not. >>>> "Much more recently it became known that one of the earliest events in the course of TMV infection is the presentation of a viral 5'-terminal guanosine (known as cap0) to the infected cell. This terminal structure  is monomethylated in the N7 position and comonly encountered attached via an  inverted 5'-5' linkage to the genomic positive-sense RNA of the virus. When exposed, this structure may become available to interact with cellular components (including ribosomes) before complete uncoating of the virions.   page 02 - line 96: Please correct "coronavirises"   page 03 - line 138: "...and motifs V-VIII had roles in interacting with both SAM and the substrate (ref)". Please, insert the reference accordingly.   page 04 - line 184: Please, include the parameters used to search for motives in programs such as PSI-BLAST and HHpred.   page 10 - line 456: Please, change "most commonly and aspartic.." for "most commonly an aspartic...".   page 13 - line 604: "more than 6000 virus RNA-directed RNA polymerases (ref)". Please, insert the reference accordingly.   page 13 - line 608: Please, correct "counteclockwise" for "counterclockwise"   page 13 - line 609: I believe "blue, Branch II" should be changed to "blue, Branch III"   Figure 3: This figure needs to be reworked. Why use the the same code of colors for the inner and outer circle if they mean a different thing? Also, the quality of the image is difficulting to see the names of some viral groups, especially the light-colored ones. I suggest to increase the quality of this figure and also choose another set of colors for it. Light blue and light grey are very much alike. 

Author Response

page 01 - line 31: Please change "later" for "then". 
---Done

page 01 - line 32-35: "Much more recently it became known that one of the earliest events in the course of TMV infection is presentation to an infected cell of the 5'-terminal guanosine, which is monomethylated in the N7 position and attached via the inverted 5'-5' linkage to the genomic positive-sense RNA of the virus. This terminal structure, known as cap0, may become exposed and available for interaction with the cell, including ribosomes, before complete uncoating of the virions". I would suggest a slight modification in the sentence for the sake of comprehension . Feel free to change it or not. 
"Much more recently it became known that one of the earliest events in the course of TMV infection is the presentation of a viral 5'-terminal guanosine (known as cap0) to the infected cell. This terminal structure  is monomethylated in the N7 position and comonly encountered attached via an  inverted 5'-5' linkage to the genomic positive-sense RNA of the virus. When exposed, this structure may become available to interact with cellular components (including ribosomes) before complete uncoating of the virions.  
---Done

page 02 - line 96: Please correct "coronavirises"  
--- Done

page 03 - line 138: "...and motifs V-VIII had roles in interacting with both SAM and the substrate (ref)". Please, insert the reference accordingly.  
--- Done

page 04 - line 184: Please, include the parameters used to search for motives in programs such as PSI-BLAST and HHpred.  
--- Done

page 10 - line 456: Please, change "most commonly and aspartic.." for "most commonly an aspartic...".  
--- Done

page 13 - line 604: "more than 6000 virus RNA-directed RNA polymerases (ref)". Please, insert the reference accordingly.  
--- Done

page 13 - line 608: Please, correct "counteclockwise" for "counterclockwise" 
--- Done

page 13 - line 609: I believe "blue, Branch II" should be changed to "blue, Branch III"  
--- Done

Figure 3: This figure needs to be reworked. Why use the the same code of colors for the inner and outer circle if they mean a different thing? Also, the quality of the image is difficulting to see the names of some viral groups, especially the light-colored ones. I suggest to increase the quality of this figure and also choose another set of colors for it. Light blue and light grey are very much alike.  
--- The tree is relabeled and the legend is changed. 

Reviewer 2 Report

This manuscript is a comprehensive review of the enzymes, capable of methylating RNA, that are encoded by RNA viruses. It takes a broad view of this interesting topic and is a valuable addition to the literature. I have only a few very minor criticisms.

1.P. 3, line 138, has a “ref” where apparently a citation belongs.

2.P. 10, line 458, refers to vesicular stomatitis virus but cites a paper on Sendai virus. (In fact the comment undoubtedly applies to both viruses, but this discrepancy in the text should be corrected).

3.Figure 2 is too small and too dark to be useful. The legend mentions numbering but I could not discern any numbers in the figure.

Author Response

This manuscript is a comprehensive review of the enzymes, capable of methylating RNA, that are encoded by RNA viruses. It takes a broad view of this interesting topic and is a valuable addition to the literature. I have only a few very minor criticisms.
1.P. 3, line 138, has a “ref” where apparently a citation belongs. 
--- Corrected
2.P. 10, line 458, refers to vesicular stomatitis virus but cites a paper on Sendai virus. (In fact the comment undoubtedly applies to both viruses, but this discrepancy in the text should be corrected). 
--- Corrected
3.Figure 2 is too small and too dark to be useful. The legend mentions numbering but I could not discern any numbers in the figure.  
--- This must be the artifact of conversion at the Editorial Office; the pdf version that I was able to generate is fully zoomable and the numbers are clearly seen. I attach a high-resolution source with this response.

Reviewer 3 Report

This manuscript is a very interesting study on viral methyltransferases. The work is appropriate for publication in Biomolecules. Please find below some comments for improvement.

Comments:

Overall: I think addition of more figures would be helpful to the reader; e.g. Rossmann folds in the introduction, flow charts in materials and methods, schematics in each result section, etc.

Introduction: I felt the introduction was quite long - could some of the information be condensed or moved into the discussion? E.g. the paragraph on cap0, cap1 and cap2 methylations.

Lines 30-31: Maybe give the reader some sense of the timeline - when did Ivanovsky discover TMV?

Line 50: Viron???

Line 63: "from positive-strand RNA viruses SUCH as ..."

Lines 71-71: Notwithstanding viruses, which organisms harbor FtsJ/RrmJ-like methyltransferases?

Line 191: Italicise Riboviria.

Lines 130, 158: What does "in the following" mean?

Materials and Methods: How were the representative members of each family selected? Was it a random selection among full length sequences or was a clustering algorithm used?

Results: My main concern in the results is the proposed evolutionary scenario (lines 648-661). Although I do not dispute that the methyltransferase domains in Riboviria are most likely monophyletic, I am not convinced that such domains were not introduced multiple times to different viruses through their hosts. A more wide phylogenetic study including eukaryotic methyltransferase domains in addition to viral ones would be required to settle this one way or another ...

Line 604: Missing reference?

Line 609: There are two 'Branch II' and no 'Branch III'.

Lines 723-726: Fungi recognise dsRNA molecules (i.e. viral genomes and viral replication intermediates) as part of their antiviral RNA silencing mechanisms.

Author Response

Overall: I think addition of more figures would be helpful to the reader; e.g. Rossmann folds in the introduction, flow charts in materials and methods, schematics in each result section, etc.
--- After a consideration, I respectfully decline this suggestion. Excellent illustrations of Rossmann folds are easy to find online (a dedicated Wikipedia entry and beyond), and the flowcharts would in my opinion be a distraction from the novel information presented here. I suspect that many readers would be interested in Figure 1 only :)

Introduction: I felt the introduction was quite long - could some of the information be condensed or moved into the discussion? E.g. the paragraph on cap0, cap1 and cap2 methylations. 
--- As for that specific paragraph, cap2 and beyond may be less relevant, but the discussion of cap0 vs cap1 becomes important later in the paper. However, I have deleted inthree long sentences from the Introduction to make it slightly shorter. Some of the extra length of the Introduction is due I tried to a little historical perspective, which I think is appropriate given that this is a contribution to a history-themed volume (“Virology 130 years after Ivanovsky”). 

Lines 30-31: Maybe give the reader some sense of the timeline - when did Ivanovsky discover TMV? 
--- Done

Line 50: Viron??? 
--- Corrected

Line 63: "from positive-strand RNA viruses SUCH as …" 
--- Corrected

Lines 71-71: Notwithstanding viruses, which organisms harbor FtsJ/RrmJ-like methyltransferases? 
--- Information added, along with the NCBI COG numbers

Line 191: Italicise Riboviria. 
When the entire sentence / heading is italicized, it is customary to un-italicize the Latinized name for contrast. I leave it to the Editorial Office to advise on the style. 
Lines 130, 158: What does "in the following" mean?
--- Changed to “in the rest of this paper”

Materials and Methods: How were the representative members of each family selected? Was it a random selection among full length sequences or was a clustering algorithm used? 
--- For each family, a sequence represented in the Genome Division of GenBank was selected randomly, without clustering (ICTV-approved and NCBI-accepted taxonomic assignment to a family, of course, is in itself clustering). I made an edit in the first paragraph of Materials and Methods.

Results: My main concern in the results is the proposed evolutionary scenario (lines 648-661). Although I do not dispute that the methyltransferase domains in Riboviria are most likely monophyletic, I am not convinced that such domains were not introduced multiple times to different viruses through their hosts. A more wide phylogenetic study including eukaryotic methyltransferase domains in addition to viral ones would be required to settle this one way or another … 
--- I agree and have edited the Discussion to make these concerns more explicit.

Line 604: Missing reference? 
--- Corrected.

Line 609: There are two 'Branch II' and no 'Branch III'. 
--- Corrected.

Lines 723-726: Fungi recognise dsRNA molecules (i.e. viral genomes and viral replication intermediates) as part of their antiviral RNA silencing mechanisms. 
--- So they do. I do not know whether these systems are able to sense the methylation status of those dsRNAs – which is the subject of this paper – but this seems not to be far-fetched. I added s few words and some new references to the discussion.